# Data modeling analysis of GFRP tubular filled concrete column based on small sample deep meta learning method

**Tianyi Deng**[☯], **Chengqi Xue**[☯], **Gengpei Zhang**[iD]*

Electronic Information and Electrical Engineering School, Yangtze University, Jingzhou City, Hubei Province, China

☯ These authors contributed equally to this work.

* judgebill@126.com

## Abstract

The meta-learning method proposed in this paper addresses the issue of small-sample regression in the application of engineering data analysis, which is a highly promising direction for research. By integrating traditional regression models with optimization-based data augmentation from meta-learning, the proposed deep neural network demonstrates excellent performance in optimizing glass fiber reinforced plastic (GFRP) for wrapping concrete short columns. When compared with traditional regression models, such as Support Vector Regression (SVR), Gaussian Process Regression (GPR), and Radial Basis Function Neural Networks (RBFNN), the meta-learning method proposed here performs better in modeling small data samples. The success of this approach illustrates the potential of deep learning in dealing with limited amounts of data, offering new opportunities in the field of material data analysis.

## Introduction

Glass Fiber Reinforced Plastics (Glass Fiber Reinforced Plastics, GFRP) have been widely used in the field of civil engineering, especially in high-rise construction, long-span bridge engineering, marine engineering, underground engineering, and protective engineering [1–3]. GFRP has the advantages of being lightweight, high-strength, plasticity, strong corrosion resistance, fatigue resistance, good dielectric properties, and relatively low cost, making it a popular material choice. The current focus is on composite columns made of GFRP pipe-coated concrete or steel pipes. Compared with ordinary steel-reinforced concrete columns, these composite columns have superior mechanical properties, durability, seismic performance, and other characteristics [2, 3]. By leveraging the high strength and corrosion resistance of GFRP, combined with the advantages of concrete or steel pipes, more reliable and durable structural designs can be achieved, bringing new possibilities to the field of civil engineering.

Effective analysis of engineering data is crucial to improving the efficiency and quality of the engineering implementation process, hence advanced data processing technologies are highly regarded in the field of engineering [4]. As a powerful data processing tool, deep

**Data Availability Statement:** All relevant data are within the paper and its Supporting information files.

**Funding:** The author(s) received no specific funding for this work.

**Competing interests:** The authors have declared that no competing interests exist.

learning technology has shown great potential in the application of engineering data analysis [5]. However, there are relatively few studies reported on engineering data analysis systems. In recent years, engineering data analysis systems based on deep learning have performed excellently in the fields of manufacturing fault detection, mechanical system prediction, and dynamic system control [6]. These systems are capable of effectively processing large-scale engineering data and providing accurate analysis and predictive results, offering significant assistance to the engineering implementation process. Yet, despite remarkable achievements, a primary concern when implementing deep learning techniques in the field of engineering is the practical issue of data acquisition. Data collection at engineering sites may be influenced by various factors, such as environmental conditions, equipment limitations, and data quality, which could negatively impact the performance and accuracy of the deep learning model [7]. Therefore, addressing the issue of data collection is one of the keys to realizing the application of deep learning technology in the field of engineering.

The limited availability of datasets is indeed a major challenge when developing deep learning-based engineered data analysis systems. Generally, deep learning algorithms typically require large training datasets to leverage their advantages, as compared to traditional machine learning-based methods [8]. Datasets ten times the number of trainable weights in deep neural networks are often considered the minimum requirement, and the number of trainable weights may rapidly increase when dealing with unbalanced datasets. Even relatively simple deep learning architectures demand thousands of data samples for training. For instance, an architecture with two hidden layers, having 10 input dimensions for each hidden layer and over 200 trainable weights for each, necessitates correspondingly sized datasets for training [9]. However, reliable real-time data collection at engineering sites is costly and also requires a substantial workforce investment, leading to a scarcity of sufficiently large, relevant datasets to train reliable deep learning models. In response to this challenge, the following methods can be considered to mitigate the issues caused by limited datasets: 1. Data augmentation techniques [9–13]: By enhancing and expanding existing data, the dataset can be effectively scaled up. 2. Transfer learning [14–16]: Utilize models trained on other domains or tasks to reduce the need for extensive data through fine-tuning or transfer learning. 3. Synthetic data [17–20]: Employ synthetic data generation technologies to produce data consistent with real-world scenarios, thereby expanding the dataset. 4. Active learning [21–25]: Optimize the quality and scale of the dataset by intelligently selecting the most informative samples for annotation. These approaches can assist in developing and training deep learning models more efficiently when faced with limited datasets.

Processing a small number of engineering data samples based on deep learning is indeed a challenging problem, and meta-learning as a solution is attracting increasing attention. The concept of meta-learning is to allow the model to quickly adapt to new tasks through prior learning experience when only a small number of samples are available. This approach is similar to the way humans learn, that is, to quickly learn new knowledge through previous experience. In meta-learning, there are several common methods, including metric-based methods, model-based methods, and optimization-based methods. These methods perform well in dealing with few-sample learning problems and can help deep neural networks generalize better to new tasks. For example, meta-learning methods have been successfully applied in industrial fields, such as tool wear prediction. By fine-tuning a small number of data samples, the deep neural network trained by meta-learning shows good results under different cutting conditions. The successful application of this approach demonstrates the potential and value of meta learning in the engineering field [26–32].

However, the aforementioned meta-learning methods do not completely resolve issues related to data scarcity [33]. This is because the model should be meta-trained with a sufficient

number of data samples based on past experiences. In most cases, meta-learning focuses on image classification to leverage prior learning of visual representations for identifying arbitrary objects [34–37], thereby enabling the model to recognize specific objects. In the context of image classification models, deep neural networks can be meta-trained using large pre-existing image databases [38–40]. Subsequently, the meta-trained model can be utilized to address the small-sample learning problem. However, the direct application of such models is limited to unstructured image data.

Especially in engineering data analysis and processing, numerical data point regression methods are more relevant than image classification. In the field of engineering, it often involves the regression analysis of numerical data, such as predicting target variables based on conditions. Compared to image classification, there are relatively fewer studies on small-sample regression, but some optimization-based meta-learning methods have made some progress in this area. Using Model-Agnostic Meta-Learning (MAML) and the first-order meta-learning algorithm (Reptile) [41, 42], previous studies have successfully used these methods to predict sinusoidal oscillations based on a small number of data points. In these methods, deep neural networks are meta-trained with a large amount of data that contains sinusoidal signals of various magnitudes and periods. Subsequently, the trained networks can be used to predict trends in sinusoidal signals, even when only a small number of sample data points are available. This indicates that deep neural networks can successfully learn to describe sinusoidal oscillations as features of previous experience through a meta-learning framework.

The approach of this study is highly innovative, particularly when dealing with small-sample regression issues. By constructing a sufficiently large dataset using the prediction results of several existing regression models to facilitate the meta-training of the underlying deep neural network, the research provides new ideas for addressing the challenge of scarce data samples in engineering data analysis applications. In this study, deep neural networks were trained using data enhanced by regression models, enabling them to approximate the predicted results of these regression models. Then, the deep neural networks were trained using optimization-based meta-learning methods to learn from past experiences. The goal of optimization-based meta-learning is to find deep neural networks that are adapted to new tasks through a small number of data samples and to use the meta-trained network parameters as the initial conditions for the final fine-tuning process. This meta-learning strategy was tested in predicting GFRP-wrapped concrete to fill GFRP short columns, and the experimental results demonstrate its potential and effectiveness in handling small sample regression problems.

## Methods and theory

Fig 1 illustrates the process of the proposed small-sample regression method, including four steps: data preprocessing, data enhancement, meta-learning, and fine-tuning. In the first step, the training set and the test sets were divided and normalized. In the second step, enhancing the training data sample based on the genetic algorithm and the traditional regression model can solve the lack of meta-training data sets. In the third step, the generated virtual sample set is randomly selected for network training and learning. The fourth step is to fine-tune the deep meta-trained neural network, and combine the actual data samples to get the final results, so as to complete the small-sample regression task. Fig 2 details the process of each processing step.

### Data augmentation

This article proposes a genetic based data augmentation method as shown in Fig 3. The method borrows the idea of genetic cross mutation, treating each data sample as an individual

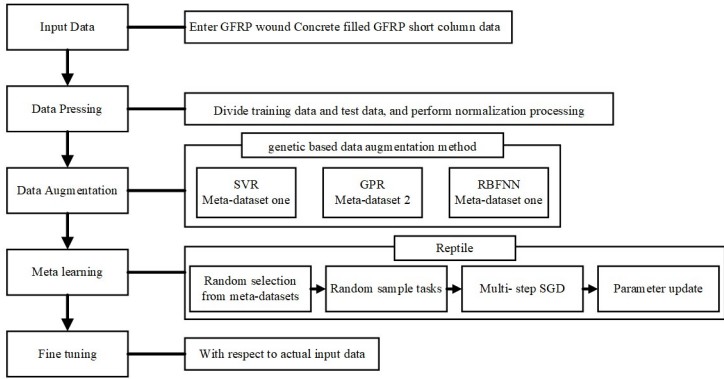

**Fig 1. Small-sample regression method.**

(1) **Data Preprocessing**
    The data of 80 percent of the total sample set were randomly divided as the training data, and the remaining 20 percent of the data were used as the test data
(2) **Data Augmentation**
    Initialize data generation model
    Input training data
    **while** not done **do**
        **for** iteration = 1,2,...**do**
            Two sets of training data were randomly sampled
            Data feature crossover and variation
            Cross-variation data is filtered by data generation model
        **end for**
    **end while**

(3) **Meta learning**
    **for** iteration = 1,2,...**do**
        Random select meta-dataset from SVR,GPR,RBFNN
        Sample task from the selected regression model
        Compute $W = U^s(\theta)$ ,denoting s steps of SGD
        Update $\theta = \theta + \beta(W - \theta)$
(4) **Fine-tuning and test**
    Adapt model parameters $\theta$ from the meta learning
    Input training data
    **for** iteration = 1,2,...**do**
        Compute gradient $\nabla_\phi L_{MSE}$ on the input data samples
        Update variable $\theta = \theta - \alpha\nabla_\phi L_{MSE}$
    **end for**
    Enter test data into the fine-tuning model
    Calculate the $R2$ and $MAPE$ of the test data

**Fig 2. Specific steps of the small-sample regression method.**

population, with parameters in the data as chromosomes. Chromosomes cross between two individuals, and by setting the probability of mutation, new data is ultimately obtained. Three models are used to evaluate fitness, that is, set the error range and screen the data within the error range to save. In the whole fitness evaluation, the three models are screened at the same

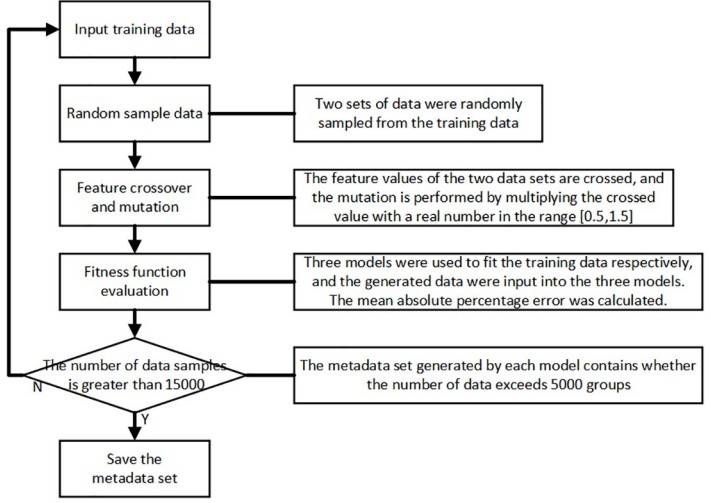

**Fig 3. Specific steps of the small-sample regression method.**

**Table 1. Characteristic of three machine learning methods.**

| | Support vector machine regression | gaussian process regression | Radial basis function neural network |
|---|---|---|---|
| *Characteristics* | 1. Maximum boundary principle: When SVM regression seeks for the regression plane or regression curve, it will try to make the boundary as wide as possible, that is, to maximize the distance between the support vector to the regression surface under the premise that the error is within a certain threshold.<br>2. Robustness: By introducing the concept called "soft interval" (soft margin), SVM regression is robust to outliers and noise and is not excessively affected by these data points.<br>3. Nonlinear mapping: SVM regression, by using kernel techniques (kernel trick), is able to handle nonlinear relationships, mapping data to a high-dimensional space for linear regression, which enables SVM to adapt to the complex data structure. | 1. Probabilistic framework: GPR provides a measure of prediction uncertainty because its prediction on a given data set is probabilistic and provides the mean and variance of the prediction.<br>2. Nonparameterization: Unlike models based on a fixed number of parameters, GPR does not rely on a pre-specified number of parameters, but deduces the shape of the function based on the similarity between the data points.<br>3. kernel function: GPR uses kernel function to define the similarity between data points. The choice of kernel function has a great impact on the performance of the model. Common kernel functions include radial basis function (RBF) and linear kernel function. | 1. Radial basis function: The core of RBFNN is its activation function, usually a Gaussian function or other radial symmetric function, used to measure the distance between the input vector and the central point.<br>2. Local approximation: Each radial basis function contributes locally to the network's output, meaning that the network is more sensitive to specific regions of the input space.<br>3. Interpolation ability: RBFNN has good interpolation ability, it can find a local model for each point in the training data, which is suitable for solving non-linear problems, especially when the distribution of data points is not uniform. |

time, and the data within the error range under the average weighted prediction of the three models will be saved. The above operations can be repeated to obtain a sufficient number of virtual samples.

This article conducted a total of seven generations of inheritance, with an initial error range of 20%. For each generation of inheritance completed, the error range decreased by 2%, and a total of 15000 sets of data were obtained, 5000 sets of virtual generated data are provided for each metadaset. The three models added include Support Vector Machines Regression (SVMR), Gaussian Process Regression (GPR), and Radial Basis Function Neural Network (RBFNN). The characteristics of these three machine learning methods are shown in Table 1. The SVMR model is widely used in the field of small sample learning. The optimized model is obtained by minimizing the total loss and maximizing the interval, which can effectively avoid overfitting and underfitting problems. The GPR model is a non parametric model that uses Gaussian process priors to analyze data, allowing for accurate fitting of targets even with very few samples. The RBFNN model can find sample center points based on clustering, reduce parameters and calculations in small sample situations, and the model training is simple and easy to implement. When there are new samples, there is no need to repeat training, just fine tune the original network to complete the task. The specific parameter settings for the three models are as follows: the kernel of SVMR adopts radial basis function, the value of parameter C is set to 100, and the other parameters are set as default parameters; The GPR adopts a fixed kernel; The number of hidden layer nodes in RBFNN is the same as the number of training data in RBFNN, and the gradient descent method is used to train the RBF network. The code of the three models is built using the third-party library Sklearn.

## Meta learning

Usually, in deep learning, a large amount of data from a certain scene is used to train the model. However, when the scene changes, the model need to be retrained. Meta learning, which means learning to learn, is represented by a child who has seen many photos of objects as they grow up. One day, when they first see a few photos of a dog, they can distinguish it from other objects.

**Table 2. The difference between meta-learning and machine learning.**

|  | Machine learning | Meta learning |
|---|---|---|
| *Objective* | Through the training data, the mapping relationship between input $X$ and output $Y$ is learned, and the function $f$ is found | Find the function F through many training tasks T and the corresponding training data D. F can output a function $f$, and f can be used for new tasks |
| *Input* | $X$ | Many training tasks and corresponding training data |
| *Function* | $f$ | $F$ |
| *Output* | $Y$ | $f$ |
| *Process* | 1. Initialize the f parameter<br>2. Input data $(X, Y)$<br>3. Calculate the loss and optimize the F parameter get: $y = f(x)$ | 1. Initialize the F parameter<br>2. Enter the training task T and the corresponding training data D, and calculate the loss optimization F parameter<br>3. Get $f = F()$, $y = f(x)$ |

The differences between meta learning and machine learning are shown in Table 2. In machine learning, the training unit is a piece of data that optimizes the model. The data can be divided into training, testing, and validation sets. Machine learning optimizes model parameters by calculating losses on the training set to obtain a model that directly solves the problem. In meta learning, the training units have hierarchical relationships. The first level of training units is the task, which means there are many problems. However, there is a premise that the problems must have similarity, which can be all images or text, but cannot be crossed. The second training unit is the data corresponding to the task. The purpose of both is to find a function, but the functions of the two functions are different and the problems they solve are different. Machine learning functions directly act on features and labels, finding the connection between features and labels. The function in meta learning is to find a new f, which is explained in detail in the meta learning methods MAML and Reptile before being used to solve problems.

To address the small-sample problem, the MAML (Model-Agnostic Meta-Learning) method samples data during training in units of tasks, with each task containing a support set and a query set. The direction of parameter updates for the model in the MAML method is shown in Fig 4 (left). Initially, the model parameters are initialized to $\theta_0$. Then, the first task $task_m$ is sampled, and the gradient is calculated on the support set within the task. The update

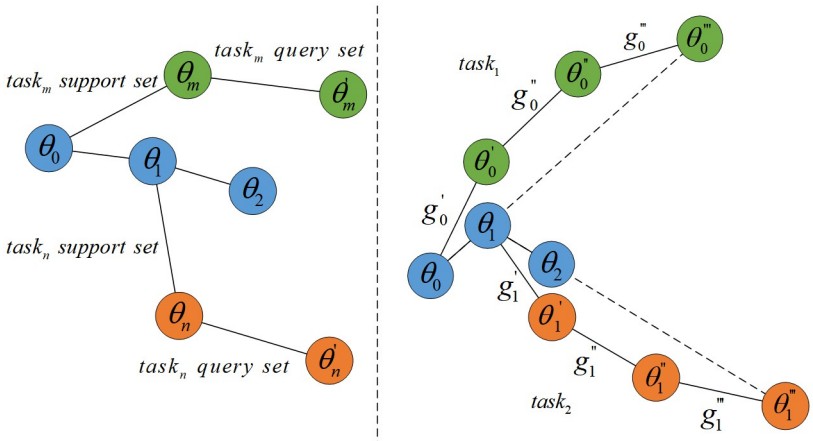

**Fig 4. The parameter update process for MAML and Reptile.**

direction for the model parameters on the task's query set is determined, which is the change in the blue parameters in Fig 4 (left). This adapts to the sampled task $task_m$ but does not fully adapt. Similarly, the second task is sampled, and the model parameters $\theta_1$ continue to be updated. After several iterations, the training process of the MAML method is completed.

Compared to MAML, the Reptile method is simpler. Training data is still sampled by taking several tasks, but there is no need to divide into support and query sets. The Reptile method directly uses the data from a single sampled task for a small number of gradient descent steps. In Fig 4 (right), three gradient descent steps are performed, with the gradients calculated for each step being $g_0'$, $g_0''$ and $g_0'''$ The single update formula for the model parameters in Reptile is $\theta_1 = \theta_0 - \beta(g_0' + g_0'' + g_0''')$, where $\beta$ is the learning rate. Following this process, several tasks are sampled to complete the model training work. During testing, it is consistent with MAML.

## Model establishment

The data comes from the research of scholars such as [30]. Investigation on various section GFRP profiles strengthening concrete-filled GFRP tubular columns studied the influence of three types of GFRP profiles on the ultimate bearing capacity of concrete filled GFRP pipe columns through axial compression test and numerical analysis. The cross-sectional shapes of the three profiles are L-type, C-type, and I-type, as shown in Fig 5. A total of 72 column samples were used in the experiment, including 24 L-type GFRP embeddings, 24 C-type GFRP embeddings, and 24 I-type GFRP embeddings. The L-type samples are shown in Fig 6. The Control variates are used in the test. Each group controls five variable, namely, GFRP profiles section

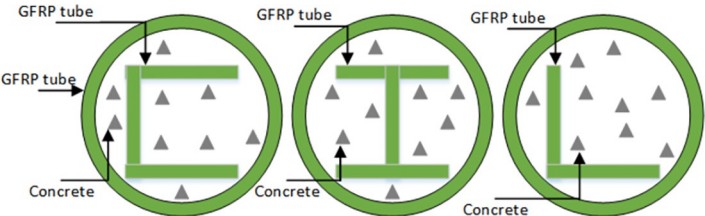

**Fig 5. The sketches of cross section.**

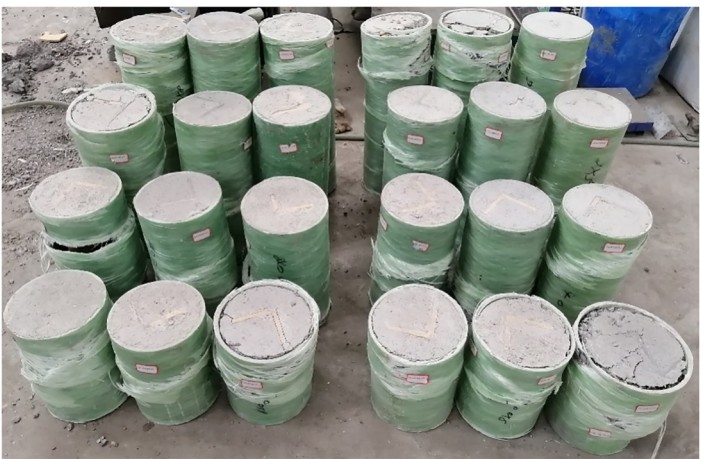

**Fig 6. Physical picture of a column.**

type, concrete strength, GFRP pipe wall thickness, GFRP pipe inner diameter, and GFRP pipe height. Taking the sample GLT6H300C20 in the data set as an example, GL represents L type, T6 represents GFRP pipe thickness 6mm, H300 represents GFRP pipe height 300mm, and C20 represents concrete strength 20MPa. The load strains curve of the column and the failure mode of the column when the test analysis variables are limited. Based on the mechanical model and experimental numerical analysis of GFRP pipe confined concrete columns, the calculation formulas for the ultimate bearing capacity of three types of GFRP profile reinforced concrete filled GFRP pipe columns were studied and summarized.

The modeling in this article is based on the two main evaluation parameters in the study: ultimate bearing capacity and ultimate displacement. The Reptile model is obtained through Engineering Few Shot Deep Reptile method learning, and the best model are trained under the other three learning methods, including SVMR, GPR, and RBFNN. Based on the above methods, prediction models for ultimate bearing capacity and ultimate displacement is established.

Regarding the input parameters for establishing the model. In order to effectively utilize all the information of the test column and describe the impact of section steel on the ultimate bearing capacity and ultimate displacement of concrete filled GFRP pipe columns, two parameters are added to the model input, namely the *SR* (Symmetrical ratio,) of the section steel and the *AR* (Area ratio,) of the section steel to the cross-sectional area of the pipe column. The calculation of the symmetry ratio is based on the shape of the section steel, and the I-steel is completely symmetrical. The symmetry ratio of I-beams is 100%, and if the angle steel is completely asymmetric, the symmetry ratio of angle steel is 0. While the channel steel is not completely symmetrical, only the proportion of the symmetrical part is calculated. The concrete strength *C* is taken as an input parameter. Secondly, because the inner diameter *D* of the GFRP tube parameters in the experiment is completely the same, in order to ensure the effective dimension of the input parameters and reduce the modeling dimension, and improve the modeling accuracy. the ratio $H/D$ of the height *H* of the GFRP tube to the inner diameter *D* of the GFRP tube, as well as the ratio $D/T$ of the inner diameter *D* of the GFRP tube to the wall thickness *T* of the GFRP tube, are taken as input parameters, and these five variables are selected as model inputs parameters. Regarding the explanation of the output parameters when establishing the model, this article selects the ultimate bearing capacity and the ultimate displacement to determine the performance of the pipe column. The greater the ultimate bearing capacity, the greater the weight that the pipe column can bear; The smaller the ultimate displacement, the higher the stability of the pipe column.

Regarding the explanation of the network structure used by the method Reptile, Reptile uses a fully connected network structure, as shown in Fig 7, which includes a five node input layer, three hidden layers, with hidden layer node sizes of 8, 16, 32, and an one node linear output layers. Batch normalization and nonlinear function Relu activation processing are used between hidden layers to prevent overfitting. Due to the low number of network layers used in deep learning research in the field of building structures, and the fact that deep learning has 5 or more layers of neural networks, combined with the fact that table data features are simpler than image data features, this paper uses a five layer fully connected network structure.

When conducting data prediction modeling, 6 sets of samples were randomly selected from 72 sets of total data to form a test set. The 6 sets of samples were composed of 2 sets of L-type samples, 2 sets of C-type samples, and 2 sets of I-type samples. The remaining 66 sets of samples formed the training set. Using methods such as Reptile, SVMR, GPR, and RBFNN to model the ultimate bearing capacity and ultimate displacement together. Then, two indicators, the coefficient of determination $R2$ and the mean absolute percentage error *MAPE*, were used

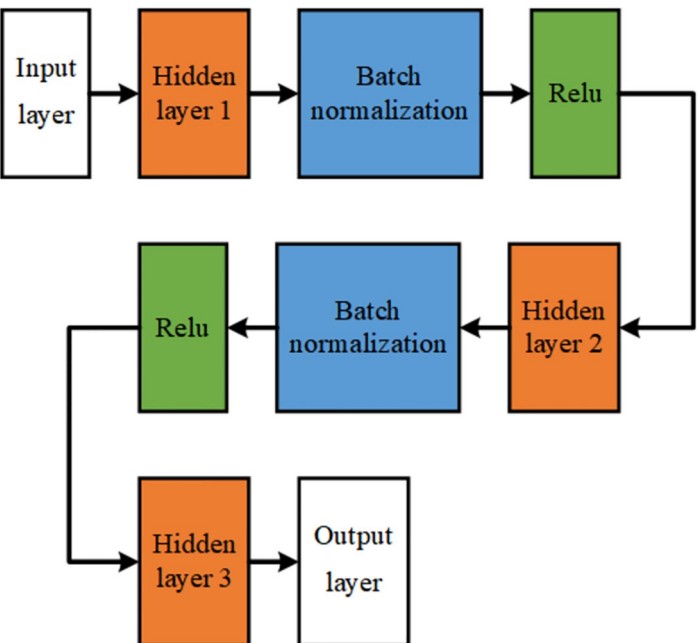

**Fig 7. Establish the network structure diagram used by the model.**

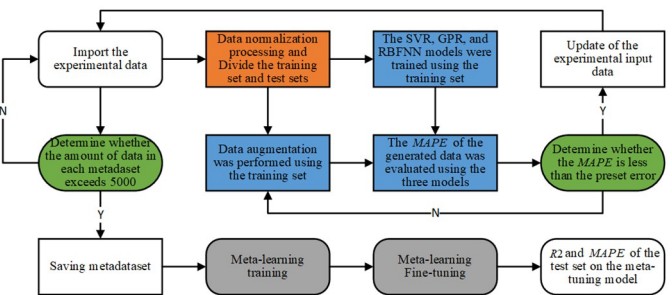

**Fig 8. Modeling process.**

to evaluate the effectiveness of the model testing. The process of establishing the model in this article is shown in Fig 8.

The specific process is described as follows:

1. Import the experimental data;

2. Data normalization processing and divide training set and testing set;

3. The SVR, GPR, and RBFNN models were trained using the training set, and data augmentation was performed using the training set;

4. The *MAPE* of the generated data was evaluated using the three models;

5. Determine whether the *MAPE* is less than the preset error, For example, the first error is 20%, if it is less than 20%, save the virtual sample data generated in the third step, and

completely replace the initial experimental input data; if it is more than 20%, return to the data enhancement process in step (3);

6. Determine whether the amount of data in each meta-dataset exceeds 5000, If the number of metadata set is greater than 5000, the metadata set is saved, and if it is less than 5000, return to the first step;

7. The metadaset was trained using the Reptile algorithm to perform the third step in Fig 2;

8. Use the training data to fine-tune the meta-learning model trained by Reptile;

9. $R2$ and $MAPE$ of the test samples were computed using the fine-tuned meta-learning model

## Model test and prediction performance analysis

The model testing process is as follows:

1. Test data input;

2. Normalization of test data: Calculate the normalization result of test data based on the normalization of training data;

3. Normalized test data is input into the model, and the model output is subjected to denormalization processing;

4. Calculate the effectiveness of model testing, calculate the coefficient of determination $R2$ and the average absolute percentage error $MAPE$.

The number of model test is 100 times, the $R2$ coefficient of the limit bearing capacity of each model is shown in Fig 9, and the $R2$ coefficient of the limit displacement of each model is shown in Fig 10. The model test results are shown in Table 3. After testing the data modeling of GFRP wrapped concrete columns, it is found that the proposed Reptile is more accurate than the three models trained with the original data. In predicting the ultimate bearing capacity of GFRP wrapped concrete columns, the three types of models have good prediction results, with the determination coefficient $R2$ calculated higher than 0.8. The maximum determination coefficient $R2$ of the Reptile model is 0.982, and the average absolute percentage error $MAPE$ of prediction is 9.5%. In predicting the ultimate displacement of GFRP wrapped concrete columns, the order of determination coefficients is: Reptile>RBFNN>SVMR>GPR, where the prediction determination coefficient $R2$ of Reptile is 0.955, and the average absolute percentage error $MAPE$ of prediction is 6.1%. In order to further compares the impact of data

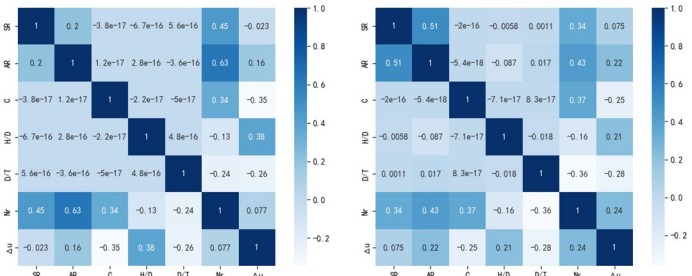

**Fig 9. The R2 coefficient of the predicted ultimate bearing capacity of each model.**

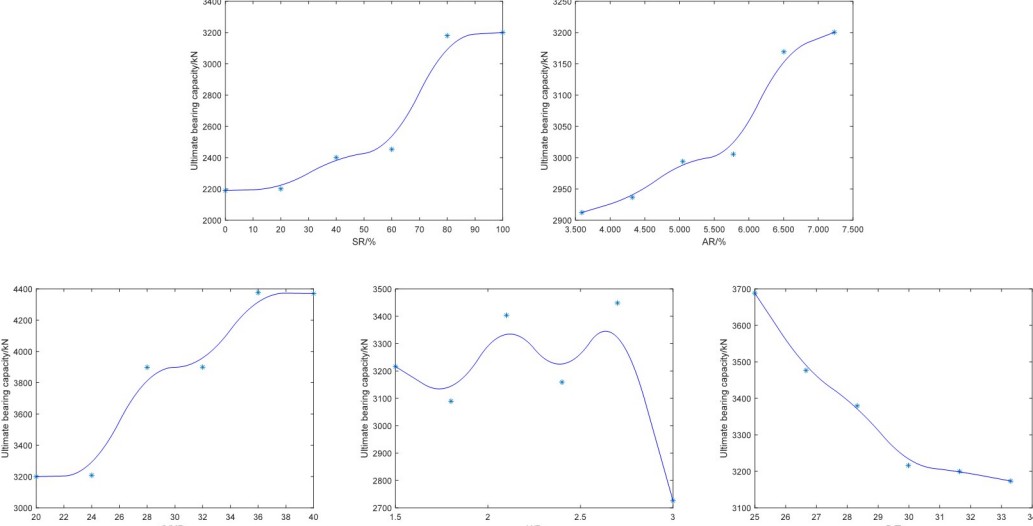

**Fig 10. The R2 coefficient of the predicted ultimate displacement for each model.**

augmentation on modeling and prediction performance in Reptile, methods such as SVMR, GPR, RBFNN, etc. Be used to model the enhanced data. The prediction performance of the model on the test data is shown in Table 3. The prediction effect of the three models using enhanced data modeling has improved, but the prediction effect of the three models using enhanced data training is still worse than that of Reptile.

By correlation analysis of the input and output parameters of the data, the role of data-enhanced modeling can be further observed. As shown in Fig 11 (left), before data enhancement, it can be seen through correlation analysis that input parameters such as symmetry ratio *SR*, area ratio *AR* and concrete strength are positively correlated with the ultimate load Nr of output parameters, while *H/D* and *D/T* are negatively correlated with the ultimate load. Through data enhancement, the number of samples was increased, and the correlation analysis of the obtained enhanced data was shown in Fig 11 (right). The correlation of input parameters and output parameters of the enhanced data was basically consistent with that of the unenhanced data, indicating that data enhancement achieved the purpose of increasing the number of samples without changing the data distribution.

**Table 3. The difference between meta-learning and machine learning.**

| | Ultimate bearing capacity | | Ultimate displacement | |
|---|---|---|---|---|
| | **MAPE** | **R2** | **MAPE** | **R2** |
| *SVR* | 8.10% | 0.846 | 5.80% | 0.738 |
| *GPR* | 8.20% | 0.857 | 14.40% | 0.648 |
| *RBFNN* | 12% | 0.829 | 9.90% | 0.852 |
| *SVR Data Augmentation* | 8.50% | 0.866 | 7.80% | 0.772 |
| *GPR Data Augmentation* | 9.50% | 0.843 | 11.20% | 0.759 |
| *RBFNN Data Augmentation* | 10.60% | 0.85 | 7.90% | 0.875 |
| *Reptile* | 9.50% | 0.982 | 6.10% | 0.955 |

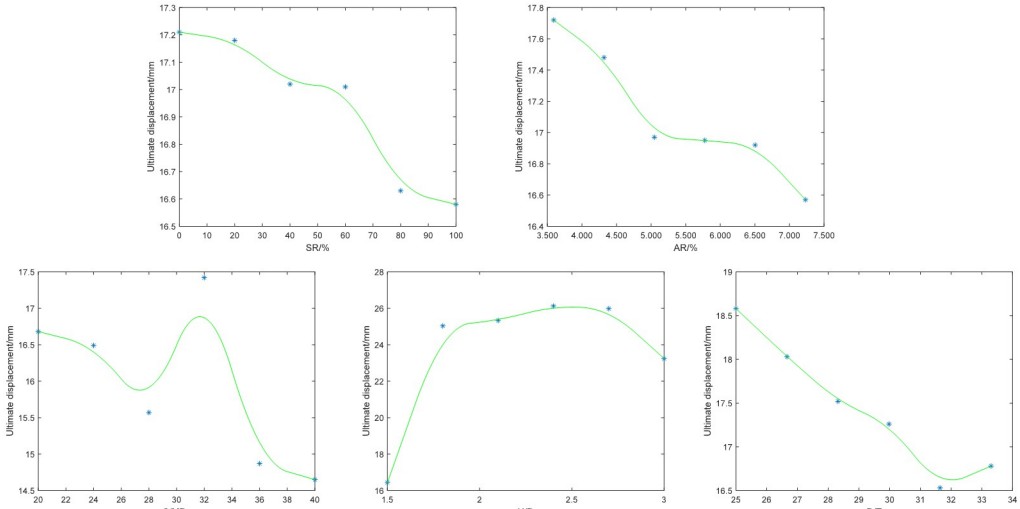

**Fig 11. Input / output correlation coefficient analysis before data enhancement (left) and after data enhancement (right).**

Data augmentation can improve the prediction performance of models such as SVMR, GPR, and RBFNN, but the final prediction performance of the three models is not as good as that of Reptile.

## Results and discussion

In order to determine the specific impact of input parameters on output parameters, the change pattern of corresponding output parameters under parameter changes is determined through discussion. Perform proportional interpolation on the determined parameters within the range of maximum and minimum values within the original data. For example, perform pentagonal interpolation to obtain a data table consisting of six sets of input data. Input this table data into the Reptile model in Section 4, draw a curve based on the model output, and analyze the impact of parameter changes on output parameter changes based on the curve.

### Ultimate bearing capacity analysis

As shown in Fig 12, the ultimate bearing capacity of the column shows an upward trend with the increase of symmetry ratio. When the symmetry ratio exceeds 80%, the ultimate bearing capacity of the column remains almost unchanged. The trend of the influence of area ratio on the ultimate bearing capacity of columns is similar to that of symmetry ratio. When the area ratio reaches its maximum, the ultimate bearing capacity of columns is the highest. As the strength of concrete increases, the ultimate bearing capacity of the column increases, which is consistent with the conclusion (2) in the study: due to the increase in concrete strength, the ultimate bearing capacity of all specimens increases. When $H/D$ increases within the interval, the variation in the ultimate bearing capacity of the column fluctuates greatly, but the overall trend is decreasing, partially in line with the conclusion (3) in the study: when other experimental parameters remain unchanged, the ultimate bearing capacity of all specimens of the three groups of columns decreases with the increase of column height. Although overall, the larger the value of $H/D$, the lower the ultimate bearing capacity of the column, it cannot be

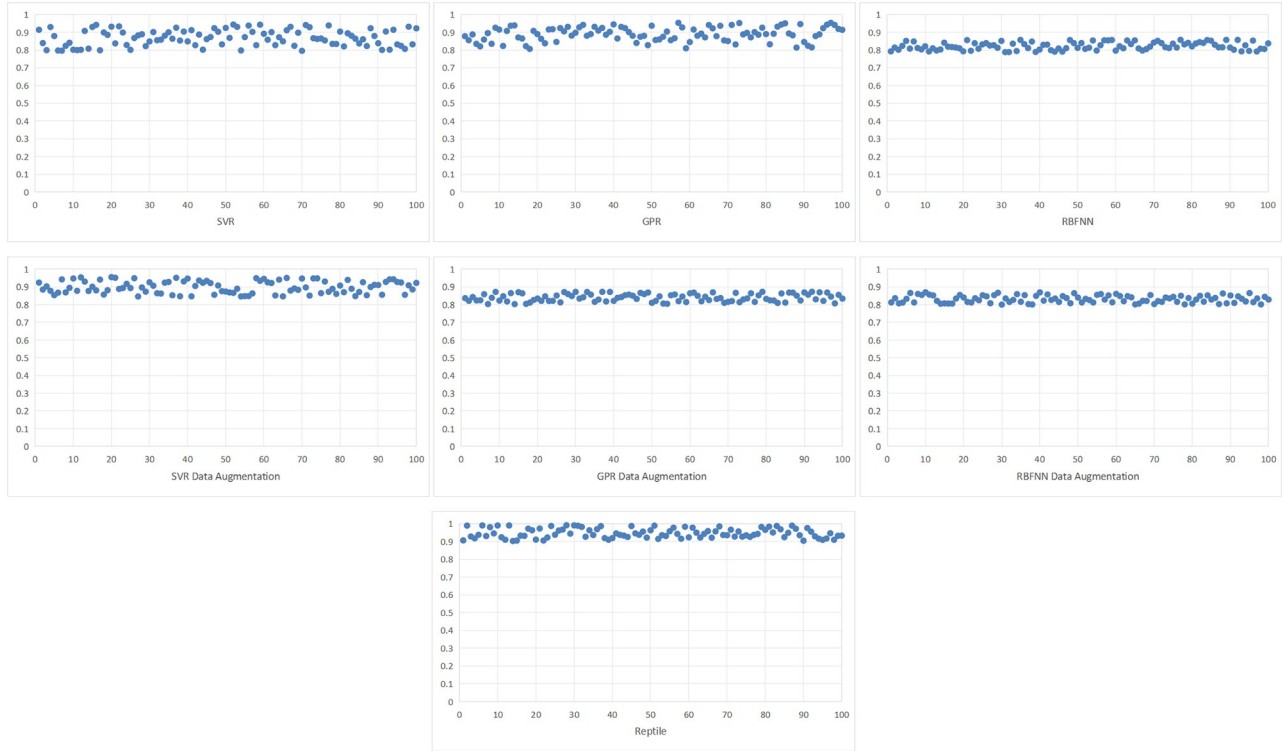

**Fig 12. The law of ultimate bearing capacity affected by each parameter.**

ignored that the alternating increase and decrease in the ultimate bearing capacity of the column occurs to the increase of $H/D$ values. When $D/T$ increases within the interval, the ultimate bearing capacity of the column shows a downward trend. The smaller the $D/T$, the greater the ultimate bearing capacity of the column. This is because the original experiment used the same inner diameter $D$, and the smaller the $D/T$, which is equivalent to the thickness $T$ of the GFRP pipe, which has the effect of reinforcement.

## Ultimate displacement analysis

As shown in Fig 13, the ultimate displacement of the column decreases with the increase of symmetry ratio. When the symmetry ratio exceeds 80%, the ultimate displacement remains almost unchanged. The only trend in which the area ratio affects the ultimate displacement of a column is similar to the symmetry ratio. When the area ratio reaches its maximum value of 7.232%, the ultimate displacement of the column is the smallest. Within the variation range of concrete strength, the ultimate displacement of the column first decreases to the local lowest point, then rapidly increases to the global highest point, finally rapidly decrease, and finally stabilizes. When the concrete strength are 40MPa, the ultimate displacement of the column is the smallest. Within the range of $H/D$ variation, the ultimate displacement of the column first increases, then tends to stabilize, and finally shows a downward trend. When $H/D$ is 1.5, the ultimate displacement of the column is the smallest. Within the variation range of $D/T$, the overall ultimate displacement of the column shows a downward trend, and when the $D/T$ is 31.64, the ultimate displacement of the column is the smallest. Considering that the variation

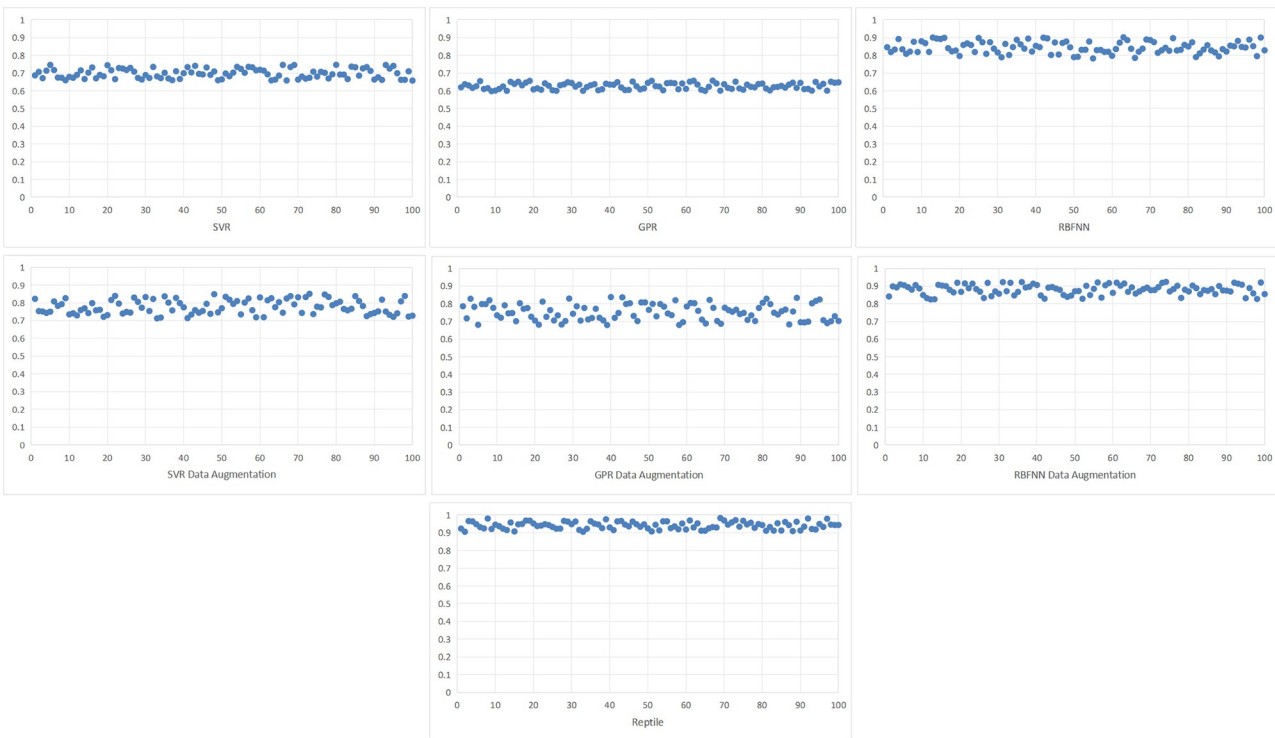

**Fig 13. The law of ultimate displacement influenced by each parameter.**

pattern of ultimate displacement was not analyzed in the original data, we attempt to supplement relevant content based on the model in this article:

This article summarizes the laws of input factors affecting the ultimate displacement of columns through the above analysis as follows:

1. The larger the symmetry ratio, the smaller the ultimate displacement of the column;

2. The larger the area ratio, the smaller the ultimate displacement of the column;

3. The greater the concrete strength, the smaller the ultimate displacement of the column. At the same time, in order to increase the ultimate bearing capacity of the column, the concrete strength can be greater than 32MPa;

4. The smaller the, the smaller the ultimate displacement of the column, but if considering the maximum possible ultimate bearing capacity, an appropriate value can be selected.

5. When the is within the range of 25 to 31.64, the larger the, the smaller the ultimate displacement of the column.

## Method outlook

This article proposes a small sample deep meta learning to model method combining multiple technical means, which can be optimized and improved in multiple aspects. When collecting more datasets related to building structures, using meta learning methods can achieve enhanced learning, further improving the model's generalization ability and

prediction accuracy. When the input dimension increases due to the change of structural unit test type, this method can be properly modified to achieve transfer learning. A larger number of data samples can generate better data augmentation evaluation criteria, which can further improve the effectiveness of data augmentation. In addition, changing the meta learning strategy is expected to further improve the training speed and prediction accuracy of the model.

## Conclusions

In this paper, a deep meta learning to model method Reptile for small sample data of building structures is proposed. This method is superior to SVMR, GPR and RBFNN in solving small sample data modeling of beams/columns.

Based on the Reptile method model testing, the following conclusions were found:

1. Updated load analysis results. It is more obvious that the contribution of $H/D$ to the ultimate bearing capacity is alternating positive and negative within the studied range. Overall, the higher the height $H$ of the pipe column, the smaller the ultimate bearing capacity of the pipe column. The contribution of $D/T$ to the ultimate bearing capacity is negative within the studied range. The smaller the thickness $T$ of the pipe wall, the smaller the ultimate bearing capacity of the pipe column.

2. Add the influence of five input factors on the ultimate displacement of the column; Among them, the overall contribution of $C$ to the ultimate displacement in the studied range is negative, but there is a positive improvement stage, where the ultimate displacement of the pipe column is relatively large. The overall contribution of $H/D$ to the ultimate displacement within the studied range is positively increasing. The higher the height H of the pipe column, the greater the ultimate displacement of the pipe column.

The modeling method in this article can be further studied from the aspects of collecting more relevant data, differences in input dimensions of different datasets, data sources, and meta learning methods.

It is necessary to acknowledge the limitations of this study, as our model has been trained on a limited number of samples and is restricted to the L-type, I-type and C-type columns mentioned in this paper. Therefore, the optimization design presented in this paper is selected from these three types of GFRP tubular columns.

## Supporting information

**S1 Dataset.**
(XLSX)

## Acknowledgments

We would like to express our gratitude to Xiaoyong Zhang, Wenyuan Kong, Yao Zhu, Yu Chen, Wentao Xie, Shaohua Han, Wenbo Zhou, Kang He and other esteemed scholars for their invaluable support in our research endeavors. The original data utilized in this study was sourced from the papers of these distinguished authors [29], with full approval granted by them. Furthermore, we wish to acknowledge that the physical diagram of the columns (Fig 6) and in our paper was captured through real-life photography courtesy of Xiaoyong Zhang and all other figures were created by us.

## Author Contributions

**Conceptualization:** Tianyi Deng, Chengqi Xue.

**Data curation:** Tianyi Deng, Chengqi Xue.

**Formal analysis:** Tianyi Deng, Chengqi Xue.

**Funding acquisition:** Tianyi Deng, Chengqi Xue.

**Investigation:** Tianyi Deng, Chengqi Xue.

**Methodology:** Tianyi Deng, Chengqi Xue, Gengpei Zhang.

**Project administration:** Tianyi Deng, Chengqi Xue.

**Resources:** Tianyi Deng, Chengqi Xue.

**Software:** Tianyi Deng, Chengqi Xue.

**Supervision:** Tianyi Deng, Chengqi Xue.

**Validation:** Tianyi Deng, Chengqi Xue.

**Visualization:** Tianyi Deng, Chengqi Xue.

**Writing – original draft:** Tianyi Deng, Chengqi Xue.

**Writing – review & editing:** Tianyi Deng, Chengqi Xue.

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
