## [Decision Letter · Decision Letter 0]

7 Feb 2024

PONE-D-23-41239Data modeling analysis of GFRP tubular filled concrete column based on small sample deep meta learning methodPLOS ONE

Dear Dr. Zhang,

Thank you for submitting your manuscript to PLOS ONE. After careful consideration, we feel that it has merit but does not fully meet PLOS ONE’s publication criteria as it currently stands. Therefore, we invite you to submit a revised version of the manuscript that addresses the points raised during the review process.

Dear Authors,

The evaluations from the peer reviewers regarding your submitted work have been duly received. Upon reviewing their feedback, it is evident that they recommend that you revise your manuscript. Therefore, the authors should consider each comment and decide on the best course of action for their research.

We look forward to receiving your revised manuscript.

Kind regards,

Shaker Qaidi

Academic Editor

PLOS ONE

5. We note that Figure 6 in your submission contain copyrighted images. All PLOS content is published under the Creative Commons Attribution License (CC BY 4.0), which means that the manuscript, images, and Supporting Information files will be freely available online, and any third party is permitted to access, download, copy, distribute, and use these materials in any way, even commercially, with proper attribution. For more information, see our copyright guidelines: http://journals.plos.org/plosone/s/licenses-and-copyright.

1. You may seek permission from the original copyright holder of Figure 6 to publish the content specifically under the CC BY 4.0 license.

Additional Editor Comments:

Reviewers' comments:

Reviewer's Responses to Questions

**Comments to the Author**

1. Is the manuscript technically sound, and do the data support the conclusions?

Reviewer #1: Yes

Reviewer #2: Partly

2. Has the statistical analysis been performed appropriately and rigorously? 

Reviewer #1: I Don't Know

Reviewer #2: N/A

3. Have the authors made all data underlying the findings in their manuscript fully available?

Reviewer #1: Yes

Reviewer #2: Yes

4. Is the manuscript presented in an intelligible fashion and written in standard English?

Reviewer #1: Yes

Reviewer #2: Yes

5. Review Comments to the Author

Reviewer #1: Review Comments to Author

Review for “Data modeling analysis of GFRP tubular filled concrete column based on small sample deep meta learning method”

Reviewer Comments:

Data modeling analysis of GFRP tubular filled concrete column based on a small sample deep meta-learning method is investigated in this study. The reviewer suggests it can be reconsidered if the authors can address the comments as follows:

1. Page 1, the manuscript in the title page (Title, Author, Affiliations, Abstract) lacks the line number, which makes it too difficult for the reviewer to locate the possible comments on this study.

2. Page 1, Author lists, Tianyi Deng1☯, Chengqi Xue1☯, Gengpei Zhang1*. What does the symbol “☯” mean for the authors? Maybe it represents the authors’ contribution equally to this research. In addition, what is this symbol “*” stands for? Corresponding author or other meanings? Considering the E-mail address is provided in the next line. Please clarify. Add the corresponding author before the E-mail address.

3. Page 1, the affiliation “, Yangtze University, Jingzhou City, Hubei Province, China”. The school of affiliation is correct? The reviewer recommends “School of Electronics and Information”. Please confirm this. The postcode is neglected in the affiliation.

4. Page 3, Line 76, Introduction, “using concrete filled GFRP column data as the test object to solve the modeling and analysis problem of small sample data in building structural units”. The abbreviation GFRP is the first time appeared in the manuscript without providing the full name. Please check the whole manuscript for similar issues.

5. The manuscript is entitled “GFRP tubular filled concrete column”. However, the reviewer could not find any information about why GFRP tubular filled concrete columns were chosen for the research object. Please explain the reasons.

6. The illustration of the Figs. 1~4, 7~8 was low resolution with poor quality. The reviewer suggests the authors redraw the flowchart by using Visio software. Special attention should be paid to the font, preferably Times New Roman.

7. Table 1, function “f and F” should be changed into “ and ”. Please check all the symbols in the whole manuscript. In the process, (X, Y), y=f(x), f=F(), please use the MathType software for the symbols, Italic Font.

8. Figs. 5 and 6 should be cited because they are not your own tested specimens.

9. Figs. 9~13 is the same style, they could combine in the same figure with sub-figure (a)~(d). Considering your expertise, please change the drawing style from Excel to Origin, Matlab, or Python. Vector graph is highly recommended in scientific graphs. Figs. 14~18 is a similar situation.

10. Page 10, Tables 2 and 3, “ultimate displacement” should be replaced by “Ultimate displacement”, which can consist of “Evaluation criteria” and “Evaluation criteria”.

11. Page 15, Lines 351~352, This method is superior to SVMR, GPR, and RBFNN in solving small sample data modeling of beams/columns. Does the proposed method consider GFRP-reinforced concrete beams? Please clarify.

12. Page 15, Lines 354~357, H/D and D/T should be replaced by “ and ”. Please check similar issues in the whole manuscript.

13. Please add DOI for the references listed.

14. Much literature has been reviewed in the paper, but less critical. The latest research concerning this topic should be addressed and added in the introduction. In other words, you should clearly explain what contribution that has been made by former research and, in particular, what limitation or weakness that exists in each previous research. You should identify the gap in the knowledge from the review of the previous research to justify the significance of YOUR current research topic.

15. Some assumptions are stated in various sections. Justifications should be provided for these assumptions. What is the basic principle of SVR, GPR, RBFNN, and SSDML? Evaluation of how they will affect the results should be made.

16. What are the concrete and GFRP material properties of the tested specimens? How to calculate the ultimate bearing capacity?

17. The authors should clarify the novelty of this manuscript.

18. The background and research needed on GFRP should be demonstrated with more words in the introduction section.

19. The authors should clarify how this method is to be used for GFRP composite structures in civil engineering.

20. The data points in this research are not quite enough. How the authors can guarantee the accuracy of this modeling approach?

21. Page 13, line 308. As shown in Fig 14 Fig 18, the ultimate displacement of the column decreases with the increase of the symmetry ratio. Figs. 14 and 18.

22. The comparison of the data is too common with only qualitative analysis, please rewrite the style with quantitative analysis.

In its current version, the reviewer recommends the document for Major Revision in PLOS ONE.

Reviewer #2: This paper proposes an innovative deep meta learning modeling approach called SSDML to address key small sample data challenges in machine learning models for civil engineering applications. The authors demonstrate empirically on a concrete-filled GFRP column dataset that SSDML outperforms other methods by leveraging data augmentation and meta-learning to improve prediction accuracy. Strengths include the novel integration of techniques, detailed experimental results, and supplemental numerical analysis enriching field knowledge. Considerable potential exists for extending this work by tuning model hyperparameters, incorporating additional datasets, and adapting the framework using transfer learning. Further model optimization and validation will help strengthen the conclusions and better quantify method advantages. Overall the study reflects substantive research contributions advancing small sample modeling capabilities, though scope remains for building out the technical evidence base and enhancing engineering interpretability. If addressed, the gaps identified should not diminish the merits of this promising approach tackling a discipline priority problem.

1. In the abstract, consider revising the language to be more concise. Some sentences could be shortened without losing key information.

2. In the introduction, provide more context and motivation early on regarding why small sample modeling is important for this application. What key challenges does it aim to address?

3. In the data augmentation section, provide more specific details on the models used and parameters selected. Were these optimized in any way? How was model performance evaluated?

4. Were any data preprocessing or feature engineering steps taken before model fitting? This could help improve accuracy.

5. For the meta learning methods, consider including a bit more background explanation of how they differ and their relative advantages.

6. In the data and modeling section, explain the rationale for the network architecture selected. Was any hyperparameter tuning conducted?

7. For the model testing results in Tables 2 and 3, include the sample sizes used. Also explain if a validation set was held out.

8. In the model testing conclusion section, specifically compare the performance lift from data augmentation vs. the final SSDML model to quantify that added impact.

9. For the discussion section analysis, consider visualizing the trends with plots for easier interpretation.

10. In the conclusion, summarize the key advantages empirically demonstrated by the SSDML approach compared to alternatives.

11. Carefully proofread the paper to fix any grammatical issues or awkward phrasing. Focus on clear, concise language.

12. Review citation formats to ensure consistency throughout.

13. Consider additional references demonstrating successful applications of meta learning and data augmentation to small sample modeling.

14. Provide more details on the specific models used as baselines (SVMR, GPR, etc.) in terms of the algorithms, libraries implementations, and parameter settings.

15. Explain any data splitting, cross-validation, or other validation strategies used to reduce overfitting and evaluate model generalizability.

16. Discuss any limitations or assumptions in the data modeling approach.

17. Elaborate on the rationale and process for selecting the specific input features used.

18. Provide more details on the genetic algorithms and parameters used for data augmentation.

19. Explain the basis for quantifying model accuracy specifically using R^2 and MAPE. Also consider additional metrics.

20. Discuss the potential for extending this approach to other types of building structure modeling tasks using transfer learning.

6. PLOS authors have the option to publish the peer review history of their article (what does this mean?). If published, this will include your full peer review and any attached files.

Reviewer #1: No

Reviewer #2: **Yes: **Mahmoud Akeed

---

## [Author Response · Author response to Decision Letter 0]

14 Mar 2024

The authors wish to thank the Editors and the anonymous reviewers very much for their valuable comments and suggestions, which leads to significant improvement of the quality and presentation of the manuscript.We revised the article according to the corresponding template, and the modified sections are highlighted in yellow. In the following, we will explain how the journal requirements and reviewer' comments have been considered in the revision. 

Reviewers' comments: 

1. Page 1, the manuscript in the title page (Title, Author, Affiliations, Abstract) lacks the line number, which makes it too difficult for the reviewer to locate the possible comments on this study. 

Author response: According to the reviewer's comments, we modified the paper format with reference to the official plos one template and added line numbers. 

2.Page 1, Author lists, Tianyi Deng1☯, Chengqi Xue1☯, Gengpei Zhang1*. What does the symbol “☯” mean for the authors? Maybe it represents the authors’ contribution equally to this research. In addition, what is this symbol “* ” stands for? Corresponding author or other meanings? Considering the E-mail address is provided in the next line. Please clarify. Add the corresponding author before the E-mail address.

Author response: According to the reviewer's suggestion and using the official template, we re-modified the labels in the author list to match the official author list template.

3.Page 1, the affiliation “, Yangtze University, Jingzhou City, Hubei Province, China”. The school of affiliation is correct? The reviewer recommends “School of Electronics and Information”. Please confirm this. The postcode is neglected in the affiliation. 

Author response: According to the reviewer's suggestion, we have revised the affiliated unit and postal code.

4. Page 3, Line 76, Introduction, “using concrete filled GFRP column data as the test object to solve the modeling and analysis problem of small sample data in building structural units”. The abbreviation GFRP is the first time appeared in the manuscript without providing the full name. Please check the whole manuscript for similar issues.

Author response: In accordance with the reviewer's comments, we have checked the full text for the first time in English and provided abbreviations

5.The manuscript is entitled “GFRP tubular filled concrete column”. However, the reviewer could not find any information about why GFRP tubular filled concrete columns were chosen for the research object. Please explain the reasons.

Author response: According to the reviewer's comments, we have added the reasons for filling short concrete columns with GFRP in the introduction.

Author response: In accordance with the reviewer's comments, we have We have add the performance description of the GFRP material to the first paragraph of the introduction as a reason for selection

6.The illustration of the Figs. 1~4, 7~8 was low resolution with poor quality. The reviewer suggests the authors redraw the flowchart by using Visio software. Special attention should be paid to the font, preferably Times New Roman.

Author response: According to the reviewer's comments, we redrew the graph that appeared in the paper and placed it at the end of the paper.

7. Table 1, function “f and F” should be changed into “ and ”. Please check all the symbols in the whole manuscript. In the process, (X, Y), y=f(x), f=F(), please use the MathType software for the symbols, Italic Font.

Author response: According to the reviewer's comments, we used the Mathtype editor to re-edit the relevant formulas

8. Figs. 5 and 6 should be cited because they are not your own tested specimens.

Author response: According to the reviewer's comments, we modified Figs 5 and Figs 6 to make them meet the application requirements.

9. Figs. 9~13 is the same style, they could combine in the same figure with sub-figure (a)~(d). Considering your expertise, please change the drawing style from Excel to Origin, Matlab, or Python. Vector graph is highly recommended in scientific graphs. Figs. 14~18 is a similar situation.

Author response: According to the reviewer's comments, we modified figs 9~13 and 14~18 into python vector graph format.

10.Page 10, Tables 2 and 3, “ultimate displacement” should be replaced by “Ultimate displacement”, which can consist of “Evaluation criteria” and “Evaluation criteria”.

Author response: According to reviewer's opinion, "ultimate displacement" has been changed to "Ultimate displacement" on Page 10, Tables 2 and 3.

11. Page 15, Lines 351~352, This method is superior to SVMR, GPR, and RBFNN in solving small sample data modeling of beams/columns. Does the proposed method consider GFRP-reinforced concrete beams? Please clarify.

Author response: According to the reviewer's opinion, in order to ensure the rigor of this paper, "the method is better than SVMR, GPR and RBFNN in solving the small sample data modeling of beams/columns" is modified to "The method is better than SVMR, GPR and RBFNN in solving the small sample data modeling of columns".

12.Page 15, Lines 354~357, H/D and D/T should be replaced by “ and ”. Please check similar issues in the whole manuscript.

Author response: Based on the reviewer's comments, we reviewed the full text and used the Mathtype editor to edit the variables that appeared in the full text

13.Please add DOI for the references listed.

Author response: According to the reviewer's opinion, we added DOI to the article cited in the full text

14. Much literature has been reviewed in the paper, but less critical. The latest research concerning this topic should be addressed and added in the introduction. In other words, you should clearly explain what contribution that has been made by former research and, in particular, what limitation or weakness that exists in each previous research. You should identify the gap in the knowledge from the review of the previous research to justify the significance of YOUR current research topic.

Author response: According to the reviewer's comments, we revised the introduction and made a critical analysis of the cited literature

15. Some assumptions are stated in various sections. Justifications should be provided for these assumptions. What is the basic principle of SVR, GPR, RBFNN, and EF-DR? Evaluation of how they will affect the results should be made.

Author response: According to the reviewer's comments, we added corresponding reasons before assumptions in each chapter, and added method principles such as SVR,GPR,RBFNN and EF-DR

16.What are the concrete and GFRP material properties of the tested specimens? How to calculate the ultimate bearing capacity?

Author response: According to the reviewer's comments, we have added the relevant introduction to the performance of GFRP materials in the introduction. As for how to calculate the ultimate bearing capacity proposed by the reviewer, our reply is that the mathematical formula for calculating the ultimate bearing capacity according to the investigation is very complicated, and finite element analysis is generally used for simulation in construction-related fields.

17.The authors should clarify the novelty of this manuscript.

Author response: The novelty of the manuscript we address in the introduction according to reviewer opinion

18.The background and research needed on GFRP should be demonstrated with more words in the introduction section.

Author response: According to the reviewer's comments, we added descriptions of GFRP materials and the research required

19.The authors should clarify how this method is to be used for GFRP composite structures in civil engineering.

Author response: According to the reviewer's comments, we explain in the introduction how the method provided can be applied to FRP composite structures in civil engineering

20. The data points in this research are not quite enough. How the authors can guarantee the accuracy of this modeling approach?

Author response: We use a variety of methods to improve the accuracy of meta-learning models, including adjusting hyperparameters and using regularization techniques. This helps increase the generalization ability of the model, making it more accurate to adapt to new tasks with small samples.

21.Page 13, line 308. As shown in Fig 14 Fig 18, the ultimate displacement of the column decreases with the increase of the symmetry ratio. Figs. 14 and 18.

Author response: We do not quite understand the question raised by the reviewer, can you please describe it again

22.The comparison of the data is too common with only qualitative analysis, please rewrite the style with quantitative analysis.

Author response: According to the suggestion of the reviewer, we performed a quantitative analysis, selected one variable for increasing equal numbers, while the other variables remained unchanged, and plotted the corresponding vector plots

---

## [Decision Letter · Decision Letter 1]

19 Apr 2024

PONE-D-23-41239R1Data modeling analysis of GFRP tubular filled concrete column based on small sample deep meta learning methodPLOS ONE

Dear Dr. Zhang,

Thank you for submitting your manuscript to PLOS ONE. After careful consideration, we feel that it has merit but does not fully meet PLOS ONE’s publication criteria as it currently stands. Therefore, we invite you to submit a revised version of the manuscript that addresses the points raised during the review process.

We look forward to receiving your revised manuscript.

Kind regards,

Dr. S. M. Anas, Ph.D.(Structural Engg.), M.Tech(Earthquake Engg.)

Academic Editor

PLOS ONE

Additional Editor Comments:

Dear Authors,

I hope this email finds you well. I am Dr. S. M. Anas, the academic editor assigned to handle the revision of your manuscript entitled "Data modeling analysis of GFRP tubular filled concrete column based on small sample deep meta learning method" [PONE-D-23-41239R1], for PLOS ONE. This manuscript has been reassigned to me as the previous editor was unable to respond to the editorial board.

Upon reviewing the feedback from the previous reviewers and the additional reviewers' recommendations, I have made a preliminary assessment of your revised manuscript. While one reviewer expressed satisfaction with your responses, another reviewer remains unsatisfied and opposes publication in its current form. Additionally, mixed recommendations and decisions were obtained from the newly invited reviewers by the previous academic editor.

Considering this feedback, I have decided to request a Major Revision of your manuscript. I kindly ask you to carefully address all the reviewers' comments, paying close attention to areas of concern raised by the reviewers who were unsatisfied with the previous version of your manuscript.

Please ensure that you provide detailed responses to each comment and make appropriate revisions to the manuscript to strengthen its scientific quality and clarity.

Once you have addressed the reviewers' comments and made the necessary revisions, please submit the revised manuscript along with a detailed response letter outlining the changes made and how you have addressed each comment.

If you have any questions or require further clarification, please do not hesitate to contact me. I look forward to receiving your revised manuscript.

Thank you for your attention to this matter.

Best regards,

Dr. S. M. Anas

Academic Editor

PLOS ONE

Reviewers' comments:

Reviewer's Responses to Questions

**Comments to the Author**

1. If the authors have adequately addressed your comments raised in a previous round of review and you feel that this manuscript is now acceptable for publication, you may indicate that here to bypass the “Comments to the Author” section, enter your conflict of interest statement in the “Confidential to Editor” section, and submit your "Accept" recommendation.

Reviewer #1: (No Response)

Reviewer #2: All comments have been addressed

Reviewer #3: All comments have been addressed

Reviewer #4: All comments have been addressed

Reviewer #5: (No Response)

Reviewer #6: (No Response)

2. Is the manuscript technically sound, and do the data support the conclusions?

Reviewer #1: Partly

Reviewer #2: Yes

Reviewer #3: Yes

Reviewer #4: Yes

Reviewer #5: Partly

Reviewer #6: No

3. Has the statistical analysis been performed appropriately and rigorously? 

Reviewer #1: I Don't Know

Reviewer #2: N/A

Reviewer #3: N/A

Reviewer #4: Yes

Reviewer #5: Yes

Reviewer #6: No

4. Have the authors made all data underlying the findings in their manuscript fully available?

Reviewer #1: (No Response)

Reviewer #2: Yes

Reviewer #3: Yes

Reviewer #4: Yes

Reviewer #5: Yes

Reviewer #6: No

5. Is the manuscript presented in an intelligible fashion and written in standard English?

Reviewer #1: Yes

Reviewer #2: Yes

Reviewer #3: Yes

Reviewer #4: Yes

Reviewer #5: Yes

Reviewer #6: No

6. Review Comments to the Author

**Reviewer #1**: The reviewer suggests it can be rejected because the authors cannot answer the comments carefully and correctly. The comments as follows:

1. The style of the author response is simple and elaborate without the real answer to the comments and how the new version is presented in the revised manuscript. No lines and pages was indicated in the response to reviewers.

2. Page 1, the affiliation “, Yangtze University, Jingzhou City, Hubei Province, China”. The postcode is neglected in the affiliation.

3. Page 13, line 308. As shown in Fig 14 Fig 18, the ultimate displacement of the column decreases with the increase of the symmetry ratio. Please change the sentence into “As shown in Figs. 14 and Fig 18, the ultimate displacement of the column decreases with the increase of the symmetry ratio”.

4. Line 217, the references style is not in comply with the style in introduction.

5. Line 392~393, no figures is shown in Fig.7. The title of Fig.7 is missing.

6. Line 315 and Line 318, R2 should be changed into by using MathType software.

7. Fig.4, the pictures of the cross section of GFRP pipe wrapped column is inappropriate, real pictures of the specimens are recommended.

8. The data source of this research is not quite clear. The material properties of the specimens are missing.

9. There is a space between the reference and words.

In its current version, the reviewer recommends the document for Rejected in PLOS ONE.

**Reviewer #2:** Now, the manuscript (Data modeling analysis of GFRP tubular filled concrete column based on small sample deep meta learning method) is ready for the next stage of the publishing process.

**Reviewer #3:** This research introduces an innovative deep meta-learning method called EF-DR for modelling small sample data of GFRP-wrapped concrete-filled GFRP tubular columns. The method outperforms traditional machine learning techniques like SVMR, GPR and RBFNN. The authors provide new insights into how various input factors influence the ultimate bearing capacity and displacement of the columns. The data augmentation and meta-learning approaches are promising for dealing with limited datasets in structural engineering. However, some aspects like the network architecture and outlook need further elaboration. Overall, the manuscript makes a valuable contribution to the field, with a few areas that could be strengthened.

Comments:

1. The genetic-based data augmentation method is intriguing. Could you provide more details on how the chromosomal crossover and mutation operations are specifically implemented?

2. On page 4, you mention that "reliable real-time data collection at the engineering site is costly". Have you considered any strategies to make the data collection process more efficient and cost-effective?

3. The statement on page 6 that "The prediction accuracy of each model for the ultimate bearing capacity and ultimate displacement of the output performance indicators of the pipe column is shown in Table 2" seems to be referring to the wrong table number. Please double check the table references.

4. How did you determine the hyperparameters for the fully connected network structure used in EF-DR (number of hidden layers, nodes per layer, etc.)? Did you experiment with different architectures?

5. The analysis of how input factors impact the ultimate bearing capacity and displacement provides useful practical insights. However, the physical mechanisms behind some of the observed trends are not fully explained, e.g. the alternating effect of H/D on bearing capacity. Further discussion on the underlying reasons would be valuable.

6. There seems to be a discrepancy between the concrete strength values mentioned in the data source description (20 MPa) and the values shown in Figures 7c and 7h (20-50 MPa). Please clarify the range of concrete strengths actually investigated.

7. For the model testing, only 6 out of 72 total samples were used. Is this test set sufficient to comprehensively evaluate the model performance? Validating the results on a larger test set, if possible, could increase confidence in the findings.

8. The outlook section provides good suggestions for future work. One additional direction to consider could be extending the approach to other types of structural components beyond columns, such as beams or walls. This would demonstrate the broader applicability of the method.

9. While the results show the superior performance of EF-DR compared to traditional methods, it would be informative to discuss any limitations or potential failure cases of the proposed approach. Under what circumstances might the model struggle to make accurate predictions?

**Reviewer #4:** Necessary modifications have been done and convincing explanations have been given in the revised paper. Consequently, my final recommendation is “accept”.

**Reviewer #5:** the authors did not implement all of the reviewer comments. I advised the authors to check every single comments carefully. The comments especially on the format of the graphs and the explanation is not fully clarify.

**Reviewer #6:** First of all:

A. None of the comments are responded to properly.

B. This is not the reviewers’ task to look up the MS and find your responses.

C. Entirely suffers from deep disorganization, format, ill-formatted figures, inconsistent text style, missed Tables (look at L392!!!!), informal structure and used terms like ‘method outlook’ instead of ‘conclusion’, obvious ill-formatted references like 20, 21, 22, 23, …, 42????? …

D. None of the updated relevant works neither presented nor critically analyzed.

Overall, this work without any doubt is REJECTED. Some of the reasons are:

1. Incredibly poor English full of flaws and linguistic syntaxes. ‘model modeling’???? entirely MUST be reworked by the help of a native agent. Overall, unreadable.

2. concrete lack of any innovative level is clear. In comparison with advanced published works the presented results are obviously ill-posed, inconsistent and too premature.

3. Doesn’t have any validation, model optimality approval based on the hyperparameter optimization… Nothing on how the overfitting, early convergence, trapping, error improvement monitoring, optimal topology, dying Relu … were treated. Just for example, how the kernel for SVMR was handled and adjusted?? Basically, which type of kernel and why???

4. Doesn’t have any certified and documented Discussion in terms of accuracy metrics, evidential analysis, solid comparison with other scholars, limitation, stability approval, uncertainty quantifications, pitfall and practical difficulties, any representative figure showing the results, physical interpretation,

5. The predictability, calibrating and sensitivity analysis should be carried out through the weight database of the trained model to show the importance of the used attributes when its optimality is confirmed. Clarify where and how the optimal weight database is stored? How it can be recalled?? Search for updating the neural network models using different sensitivity analysis methods, sensitivity analysis for neural networks, novel feature selection using sensitivity analysis…

6. REJECTED conclusion.

7. Any data analysis/data visualization in considering the selected attributes

7. PLOS authors have the option to publish the peer review history of their article (what does this mean?). If published, this will include your full peer review and any attached files.

Reviewer #1: No

Reviewer #2: No

Reviewer #3: No

Reviewer #4: No

Reviewer #5: **Yes: **ERTUG AYDIN

Reviewer #6: No

---

## [Author Response · Author response to Decision Letter 1]

14 May 2024

Author's Response To Reviewer Comments

Dear Reviewers:

Thank you very much for your comments and professional advice. These opinions help to improve academic rigor of our article. Based on your suggestion and request, we have made corrected modifications on the Revised Manuscript with Track Changes which we hope meet with approval. Furthermore, we would like to show the details as follows:

Reviewer #1: The reviewer suggests it can be rejected because the authors cannot answer the comments carefully and correctly. The comments as follows:

1. The style of the author response is simple and elaborate without the real answer to the comments and how the new version is presented in the revised manuscript. No lines and pages was indicated in the response to reviewers.

2. Page 1, the affiliation “, Yangtze University, Jingzhou City, Hubei Province, China”. The postcode is neglected in the affiliation.

3. Page 13, line 308. As shown in Fig 14 Fig 18, the ultimate displacement of the column decreases with the increase of the symmetry ratio. Please change the sentence into “As shown in Figs. 14 and Fig 18, the ultimate displacement of the column decreases with the increase of the symmetry ratio”.

4. Line 217, the references style is not in comply with the style in introduction.

5. Line 392~393, no figures is shown in Fig.7. The title of Fig.7 is missing.

6. Line 315 and Line 318, R2 should be changed into by using MathType software.

7. Fig.4, the pictures of the cross section of GFRP pipe wrapped column is inappropriate, real pictures of the specimens are recommended.

8. The data source of this research is not quite clear. The material properties of the specimens are missing.

9. There is a space between the reference and words.

The author’s answer: We appreciate it very much for your valuable suggestions, and we have made the following corrections according to your ideas.

1-We have revisited the standards for responses, reorganized the document's format, and strictly adhered to the template format throughout the text, both in figures and tables, and have also included DOI numbers in the references.

2-In the fifth line of page 1, the postal code is added and marked with a yellow background.

3-In the new modification, As shown in Fig 14 Fig 18 has been modified to As shown in Fig 13 with location in line 392

4-We have strictly checked the reference format of the full text, modified the literature format with the incorrect reference format, and marked the yellow background in the reference literature section.

5-For the pictures not shown, we are so sorry that we have changed the format and size of the images in lines 364 and 393.

6-The letters that should be displayed in the full text have been modified using Mathtype software and marked with a yellow background, with some modifications reflected in lines 303 to 308.

7-We used the real plot of the sample with the consent of the authors, specifically in line 225.

8-At the first delivery of the manuscript, we have standardized the source of data derived from this paper，Xiaoyong Zhang, Wenyuan Kong, Yao Zhu, Yu Chen,Investigation on various section GFRP profile strengthening concrete-filled GFRP tubular columns,Composite Structures,2022 ,Volume 283. https://doi.org/10.1016/j.compstruct.2021.115055.

The quotation marks midway with the first word have been replaced by spaces, specifically in the Literature Citation section.

Reviewer #2:Now, the manuscript (Data modeling analysis of GFRP tubular filled concrete column based on small sample deep meta learning method) is ready for the next stage of the publishing process.

The author’s answer: We feel great thanks for your professional review work on our article.

Reviewer #3:This research introduces an innovative deep meta-learning method called EF-DR for modelling small sample data of GFRP-wrapped concrete-filled GFRP tubular columns. The method outperforms traditional machine learning techniques like SVMR, GPR and RBFNN. The authors provide new insights into how various input factors influence the ultimate bearing capacity and displacement of the columns. The data augmentation and meta-learning approaches are promising for dealing with limited datasets in structural engineering. However, some aspects like the network architecture and outlook need further elaboration. Overall, the manuscript makes a valuable contribution to the field, with a few areas that could be strengthened.

Comments:

1. The genetic-based data augmentation method is intriguing. Could you provide more details on how the chromosomal crossover and mutation operations are specifically implemented?

2. On page 4, you mention that "reliable real-time data collection at the engineering site is costly". Have you considered any strategies to make the data collection process more efficient and cost-effective?

3. The statement on page 6 that "The prediction accuracy of each model for the ultimate bearing capacity and ultimate displacement of the output performance indicators of the pipe column is shown in Table 2" seems to be referring to the wrong table number. Please double check the table references.

4. How did you determine the hyperparameters for the fully connected network structure used in EF-DR (number of hidden layers, nodes per layer, etc.)? Did you experiment with different architectures?

5. The analysis of how input factors impact the ultimate bearing capacity and displacement provides useful practical insights. However, the physical mechanisms behind some of the observed trends are not fully explained, e.g. the alternating effect of H/D on bearing capacity. Further discussion on the underlying reasons would be valuable.

6. There seems to be a discrepancy between the concrete strength values mentioned in the data source description (20 MPa) and the values shown in Figures 7c and 7h (20-50 MPa). Please clarify the range of concrete strengths actually investigated.

7. For the model testing, only 6 out of 72 total samples were used. Is this test set sufficient to comprehensively evaluate the model performance? Validating the results on a larger test set, if possible, could increase confidence in the findings.

8. The outlook section provides good suggestions for future work. One additional direction to consider could be extending the approach to other types of structural components beyond columns, such as beams or walls. This would demonstrate the broader applicability of the method.

9. While the results show the superior performance of EF-DR compared to traditional methods, it would be informative to discuss any limitations or potential failure cases of the proposed approach. Under what circumstances might the model struggle to make accurate predictions?

The author’s answer: We feel great thanks for your professional review work on our article. According to your nice suggestions, we have made extensive corrections to our previous draft.

1-As your suggestion, based on gene way is very novel, our work in describing how to use genetic algorithm to solve the problem of data enhancement, will a data as an individual, the characteristics and labels of the data as genes, genetic algorithm for the two data feature value of the crossover and labels, in addition, we control the characteristic variation and tag variation to ensure the range of features and label changes is not very big, finally after several iterations, complete the virtual sample generation work. For more details, please see our modified Fig 3.

2-At present, more practical strategies, such as using existing resources, using existing data sources as much as possible, such as public databases, government statistics, academic research, etc., to reduce the cost of data collection. In addition, automated tools and software can also be used to collect data, which can reduce manual labor and improve efficiency.

3-We don't understand your description, but we need to clarify that on page 6, Table 2 describes the difference between meta-learning and machine learning, while the prediction accuracy of the final pipe column output performance index and limit displacement appears on page 12, and the corresponding table is Table 3.

4-We reduced the hyperparameter range of the fully connected network from a large range to a smaller range through several experiments, and after determining the smaller range, we further determined the best parameter set of the fully connected network through ablation experiments. For other model architectures, such as convolutional neural networks, we used them in several previous experiments with poor results.

5-About the complex phenomenon in the experiment, such as you provide the trend of alternating change, which in our opinion if further discussion need related knowledge, such as the force of the sample analysis, bearing capacity of the calculation formula, deeper analysis need tools as security, but now our team does not have these capabilities.

6-The concrete strength used in the literature of the data source only contains 20MPa, 30MPa and 40MPa. In the rule discussion, we conducted the concrete strength of the range from 20 to 40 MPa, which did not include the concrete strength outside the range, as shown in Fig 12, in line 364.

7-To ensure the accuracy of the experiment, using six sets of test samples and passing 100 experimental effects evaluation can meet the performance criteria of the comprehensive evaluation model. If the number of test samples of the model is continuously increased, the original sample set of the model is used as data enhancement will be reduced, and the authenticity of generating virtual samples will be greatly reduced.

8-It is theoretically feasible to broaden the scope of methods to other types of structures, but there are few data sets available at present. The typical dataset only includes the column, which is the limitation of our experiment, but the good performance of the method in column structure is also the potential to be applied to other structures.

It is necessary to acknowledge the limitations of this study, as our model has been trained on a limited number of samples and is restricted to the L-type, I-type and C-type columns mentioned in this paper. 

Reviewer #5:the authors did not implement all of the reviewer comments. I advised the authors to check every single comments carefully. The comments especially on the format of the graphs and the explanation is not fully clarify.

The author’s answer: We feel great thanks for your professional review work on our article.We have carefully read the evaluation of the figures and tables and revised them in the full text, partially shown in lines 130,132,163,203.

Reviewer #6:First of all:

A. None of the comments are responded to properly.

B. This is not the reviewers’ task to look up the MS and find your responses.

C. Entirely suffers from deep disorganization, format, ill-formatted figures, inconsistent text style, missed Tables (look at L392!!!!), informal structure and used terms like ‘method outlook’ instead of ‘conclusion’, obvious ill-formatted references like 20, 21, 22, 23, …, 42????? …

D. None of the updated relevant works neither presented nor critically analyzed.

Overall, this work without any doubt is REJECTED. Some of the reasons are:

1. Incredibly poor English full of flaws and linguistic syntaxes. ‘model modeling’???? entirely MUST be reworked by the help of a native agent. Overall, unreadable.

2. concrete lack of any innovative level is clear. In comparison with advanced published works the presented results are obviously ill-posed, inconsistent and too premature.

3. Doesn’t have any validation, model optimality approval based on the hyperparameter optimization… Nothing on how the overfitting, early convergence, trapping, error improvement monitoring, optimal topology, dying Relu … were treated. Just for example, how the kernel for SVMR was handled and adjusted?? Basically, which type of kernel and why???

4. Doesn’t have any certified and documented Discussion in terms of accuracy metrics, evidential analysis, solid comparison with other scholars, limitation, stability approval, uncertainty quantifications, pitfall and practical difficulties, any representative figure showing the results, physical interpretation,

5. The predictability, calibrating and sensitivity analysis should be carried out through the weight database of the trained model to show the importance of the used attributes when its optimality is confirmed. Clarify where and how the optimal weight database is stored? How it can be recalled?? Search for updating the neural network models using different sensitivity analysis methods, sensitivity analysis for neural networks, novel feature selection using sensitivity analysis…

6. REJECTED conclusion.

7. Any data analysis/data visualization in considering the selected attributes

The author’s answer: We feel great thanks for your professional review work on our article.According to your nice suggestions, we have made extensive corrections to our previous draft.

1-Regarding the bad grammar presented in the full text, we have modified the standard English grammar, including the yellow background in the introduction and other paragraphs in the full text using the yellow background.

2-First of all, we don't think our methods lack of innovation, the current research mainstream direction for machine learning direction, compared with the published works, we proposed the results are very short, but the study of further digging, join researchers in other areas of knowledge, can be specific analysis law in the discussion of complex graphics.

3-关于验证部分，我们使用了Sklearn库，这是一款使用简单的第三方库，关于审稿人您说的如何解决内核处理和调整，这在Sklearn库中很容易就实现，甚至使用Sklearn中提供的固定模型参数就可以适应很多问题。为此我们增加了图9，10，11以补充

4-With regard to the experimental indicators, we give three reliable experimental criteria, coefficient of determination and average absolute percentage error. The other suggestions you provide, we do not quite understand. Our research strictly implements the whole process from data enhancement to model building to model effect analysis.

5-Please redefine the problem, clear the storage location and way of the best weight database, we expressed doubts.

6-We do not quite understand the reason why you rejected the conclusion, but our research is obviously very innovative. Compared with some authors' simple method calls, we also studied the algorithm and described the whole process of a small sample data analysis, and the conclusion not only showed the recognition of previous conclusions, but also found new rules.

We were very confused about the reviewer's speech technique, for which we added data visualization to the full text, as shown in lines 330,332.

Sincerely yours,

Gengpei Zhang, PhD

Electronics and Information School

Yangtze University, Jingzhou, Hubei, 434100 China

---

## [Decision Letter · Decision Letter 2]

15 May 2024

PONE-D-23-41239R2Data Modeling Analysis of GFRP Tubular Filled Concrete Column Based on Small Sample Deep meta Learning MethodPLOS ONE

Dear Dr. Zhang,

Thank you for submitting your manuscript to PLOS ONE. After careful consideration, we feel that it has merit but does not fully meet PLOS ONE’s publication criteria as it currently stands. Therefore, we invite you to submit a revised version of the manuscript that addresses the points raised during the review process.

We look forward to receiving your revised manuscript.

Kind regards,

Dr. S. M. Anas, Ph.D.(Structural Engg.), M.Tech(Earthquake Engg.)

Academic Editor

PLOS ONE

Journal Requirements:

Additional Editor Comments:

Dear Authors,

I hope this email finds you well. I am writing to inform you about the status of your manuscript entitled "Data Modeling Analysis of GFRP Tubular Filled Concrete Column Based on Small Sample Deep meta Learning Method" [PONE-D-23-41239R2] submitted to PLOS ONE.

After careful consideration of the reviewers' comments and your responses, I'm pleased to inform you that one of the reviewers is satisfied with the revisions made to your manuscript. However, the other reviewer has provided feedback indicating that a few minor revisions are necessary.

Upon preliminary assessment, I have decided to move forward with a Minor Revision of your manuscript, subject to the approval of the editorial board. Please address the minor revisions suggested by the reviewer, and submit the revised manuscript at your earliest convenience.

Thank you for your attention to this matter. Should you have any questions or require further clarification, please do not hesitate to contact me.

Best regards,

Dr. S. M. Anas

Academic Editor

PLOS ONE

Reviewers' comments:

Reviewer's Responses to Questions

**Comments to the Author**

1. If the authors have adequately addressed your comments raised in a previous round of review and you feel that this manuscript is now acceptable for publication, you may indicate that here to bypass the “Comments to the Author” section, enter your conflict of interest statement in the “Confidential to Editor” section, and submit your "Accept" recommendation.

Reviewer #1: All comments have been addressed

Reviewer #3: All comments have been addressed

2. Is the manuscript technically sound, and do the data support the conclusions?

Reviewer #1: Yes

Reviewer #3: Yes

3. Has the statistical analysis been performed appropriately and rigorously? 

Reviewer #1: I Don't Know

Reviewer #3: Yes

4. Have the authors made all data underlying the findings in their manuscript fully available?

Reviewer #1: Yes

Reviewer #3: Yes

5. Is the manuscript presented in an intelligible fashion and written in standard English?

Reviewer #1: Yes

Reviewer #3: Yes

6. Review Comments to the Author

Reviewer #1: Review for “Data modeling analysis of GFRP tubular filled concrete column based on small sample deep meta learning method”

Reviewer Comments:

The authors have answered most of the comments. The additional comments are listed as follows:

1. Regarding the institution, the name of the school was not correct according to the official website. School of Electronic Information and Electrical Engineering.

2. In the supplementary Excel file “GFRP-Data”. The amount of the data was very small.

3. Page 1, Line 20, “Glass fiber reinforced polymer composites (Glass Fiber Reinforced Plastics, GFRP) have…”

4. The title of the figure should be displayed below the figure, not above the figure.

5. In Fig.5, please indicate the dimensions of the cross-sections. Check how to draw the cross-section of GFRP concrete-filled columns from the literature.

6. Please redraw Figs. 9 and 10 by using Origin software to fulfill the high-resolution requirement of scientific articles.

7. Reference 42 is empty.

8. The response to Reviewer#4 is missing.

9. The author responds to reviewer#6, comment 3. Please pay attention to the English language.

10. Page 1, Lines 16~18, Abstract, “The success of this approach illustrates the potential of deep learning in dealing with limited amounts of data, offering new opportunities in the field of engineering data analysis.” The field of engineering data analysis is a huge concept, please narrow the scale of this approach.

Reviewer #3: The manuscript (Data Modeling Analysis of GFRP Tubular Filled Concrete Column Based on Small Sample Deep meta Learning Method) has been well-revised and is ready to move on to the next phase of publishing.

7. PLOS authors have the option to publish the peer review history of their article (what does this mean?). If published, this will include your full peer review and any attached files.

Reviewer #1: No

Reviewer #3: No

---

## [Author Response · Author response to Decision Letter 2]

21 May 2024

Author's Response To Reviewer Comments

Dear Reviewers:

Thank you very much for your comments and professional advice. These opinions help to improve academic rigor of our article. Based on your suggestion and request, we have made corrected modifications on the Revised Manuscript with Track Changes which we hope meet with approval. Furthermore, we would like to show the details as follows:

Reviewer #1: Review for “Data modeling analysis of GFRP tubular filled concrete column based on small sample deep meta learning method”

Reviewer Comments:

The authors have answered most of the comments. The additional comments are listed as follows:

1. Regarding the institution, the name of the school was not correct according to the official website. School of Electronic Information and Electrical Engineering.

2. In the supplementary Excel file “GFRP-Data”. The amount of the data was very small.

3. Page 1, Line 20, “Glass fiber reinforced polymer composites (Glass Fiber Reinforced Plastics, GFRP) have…”

4. The title of the figure should be displayed below the figure, not above the figure.

5. In Fig.5, please indicate the dimensions of the cross-sections. Check how to draw the cross-section of GFRP concrete-filled columns from the literature.

6. Please redraw Figs. 9 and 10 by using Origin software to fulfill the high-resolution requirement of scientific articles.

7. Reference 42 is empty.

8. The response to Reviewer#4 is missing.

9. The author responds to reviewer#6, comment 3. Please pay attention to the English language.

10. Page 1, Lines 16~18, Abstract, “The success of this approach illustrates the potential of deep learning in dealing with limited amounts of data, offering new opportunities in the field of engineering data analysis.” The field of engineering data analysis is a huge concept, please narrow the scale of this approach.

The author’s answer: We appreciate it very much for your valuable suggestions, and we have made the following corrections according to your ideas.

1-We have changed the name of the college to keep the official name and marked with a yellow background on the fourth row.

2-The supplementary Excel files are the real real trial record data used in the study, and we also supplement the virtual sample sets we obtained in our study using the data augmentation method, and the number of virtual sample sets is about 15,000.

3-Many thanks for pointing out this error, we have modified Glass fiber reinforced polymer composites to Glass Fiber Reinforced Plastics.It's on line 20.

4-The titles of all the figures appearing in the text have been shown below the figure, partly in rows 130,132 and 163.

5-We have redrawn Figure 5 and marked the dimensions of the section. It's on line 221.In the literature, we did not find the tools in the literature to draw Fig 5, and Fig 5 in our text was drawn using the Visio software.

6-Figs 9 and 10 have been redrawn to ensure a high resolution of the images. The specific location is on lines 328 and 330.

7-We have added reference 42 at line 573。

8-Very sorry, this should not have happened. Reviewer #4 said this: Necessary modifications have been done and convincing explanations have been given in the revised paper. Consequently, my final recommendation is “accept”.Our response is that we appreciate your recognition and professional comments on our research work.

9-We are very sorry, and we will pay attention to whether our reply language meets the official requirements.

We have carefully considered the application field of the research, and in order to represent the specific application scenario of the method, the field of engineering data analysis is modified to the field of material data analysis. It's on line 18

Reviewer #3: The manuscript (Data Modeling Analysis of GFRP Tubular Filled Concrete Column Based on Small Sample Deep meta Learning Method) has been well-revised and is ready to move on to the next phase of publishing.

The author’s answer: We feel great thanks for your professional review work on our article.

Sincerely yours,

Gengpei Zhang, PhD

Electronic Information and Electrical Engineering School

Yangtze University, Jingzhou, Hubei, 434100 China

---

## [Decision Letter · Decision Letter 3]

23 May 2024

Data Modeling Analysis of GFRP Tubular Filled Concrete Column Based on Small Sample Deep meta Learning Method

PONE-D-23-41239R3

Dear Dr. Zhang,

We’re pleased to inform you that your manuscript has been judged scientifically suitable for publication and will be formally accepted for publication once it meets all outstanding technical requirements.

Kind regards,

Dr. S. M. Anas, Ph.D.(Structural Engg.), M.Tech(Earthquake Engg.)

Academic Editor

PLOS ONE

Additional Editor Comments (optional):

Dear Authors,

I am pleased to inform you that your manuscript entitled "Data Modeling Analysis of GFRP Tubular Filled Concrete Column Based on Small Sample Deep Meta Learning Method" [PONE-D-23-41239R3] has been sent to the previous reviewer for reevaluation. The reviewer is now fully satisfied with your responses and has recommended the paper for publication.

Based on the reviewer's comments and a preliminary assessment of the revised manuscript, I have decided to accept your manuscript, subject to the approval of the editorial board.

The journal office will be in touch with you soon regarding the further steps for the publication process.

Congratulations on your accomplishment, and thank you for choosing PLOS ONE as your publication venue.

Best regards,

Dr. S. M. Anas

Academic Editor

PLOS ONE

Reviewers' comments:

Reviewer's Responses to Questions

**Comments to the Author**

1. If the authors have adequately addressed your comments raised in a previous round of review and you feel that this manuscript is now acceptable for publication, you may indicate that here to bypass the “Comments to the Author” section, enter your conflict of interest statement in the “Confidential to Editor” section, and submit your "Accept" recommendation.

Reviewer #1: All comments have been addressed

2. Is the manuscript technically sound, and do the data support the conclusions?

Reviewer #1: Yes

3. Has the statistical analysis been performed appropriately and rigorously? 

Reviewer #1: Yes

4. Have the authors made all data underlying the findings in their manuscript fully available?

Reviewer #1: Yes

5. Is the manuscript presented in an intelligible fashion and written in standard English?

Reviewer #1: Yes

6. Review Comments to the Author

Reviewer #1: (No Response)

7. PLOS authors have the option to publish the peer review history of their article (what does this mean?). If published, this will include your full peer review and any attached files.

Reviewer #1: No

---

## [Editor Report · Acceptance letter]

5 Jun 2024

PONE-D-23-41239R3 

PLOS ONE

Dear Dr. Zhang, 

I'm pleased to inform you that your manuscript has been deemed suitable for publication in PLOS ONE. Congratulations! Your manuscript is now being handed over to our production team.

Kind regards, 

on behalf of

Dr. S. M. Anas 

Academic Editor

PLOS ONE